# Visual-Locomotion: Learning to Walk on Complex Terrains with Vision

**Wenhao Yu**[1], **Deepali Jain**[1], **Alejandro Escontrela**[1,2], **Atil Iscen**[1],
**Peng Xu**[1], **Erwin Coumans**[1], **Sehoon Ha**[1,3], **Jie Tan**[1], **and Tingnan Zhang**[1]

[1] Robotics at Google, United States
[2] University of California, Berkeley, United States
[3] Georgia Institute of Technology, United States
Email: {magicmelon, jaindeepali, aescontrela, atil, pengxu,
erwincoumans, sehoonha, jietan, tingnan}@google.com

**Abstract:** Vision is one of the essential perception modalities for legged robots to safely and efficiently navigate uneven terrains, such as stairs and stepping stones. However, training robots to effectively understand high-dimensional visual input for locomotion is a challenging problem. In this work, we propose a framework to train a vision-based locomotion controller which enables a quadrupedal robot to traverse uneven environments. The key idea is to introduce a hierarchical structure with a high-level vision policy and a low-level motion controller. The high-level vision policy takes as inputs the perceived vision signals as well as robot states and outputs the desired footholds and base movement of the robot. These are then realized by the low level motion controller composed of a position controller for swing legs and a MPC-based torque controller for stance legs. We train the vision policy using Deep Reinforcement Learning and demonstrate our approach on a variety of uneven environments such as randomly placed stepping stones, quincuncial piles, stairs, and moving platforms. We also validate our method on a real robot to walk over a series of gaps and climbing up a platform.

**Keywords:** Legged Robot, Reinforcement Learning, Visual Locomotion

## 1 Introduction

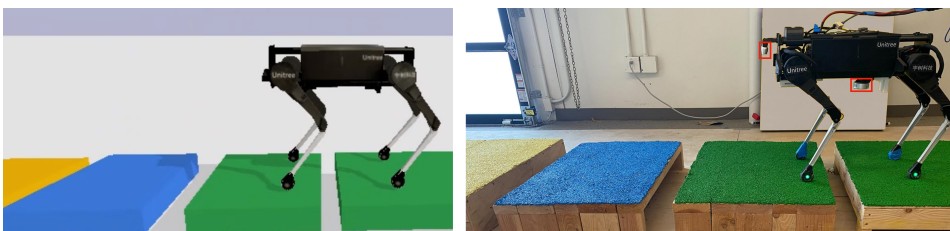

Figure 1: Our approach trains a visual-locomotion policy for the Laikago robot in simulation (left) and transfers to the real world (right). We use two depth sensors on the robot, as marked in red.

Reproducing natural vision-based locomotion skills seen in animals on artificial creatures such as legged robots has long been at the forefront of robotics research. Progress towards developing robust visual locomotion controllers not only deepens our understanding of how humans and animals perceive the environment and control our limbs, but also enables us to build autonomous machines that can reliably traverse real-world environments.

To tackle this problem, most existing methods adopt a three-stage pipeline [1, 2, 3, 4, 5, 6, 7]: perception, motion planning, and control. In the perception stage, raw sensor data such as RGBD image and/or LiDAR point clouds are carefully fused with proprioceptive streams such as IMU, motor angles, and wheel odometry. For mobile robots, a SLAM (simultaneous localization and mapping) is

5th Conference on Robot Learning (CoRL 2021), London, UK.

often used to produce a elevation terrain map centered around the robot [8]. The generated elevation map is then fed to the downstream motion planning modules to select paths, motion style, and foot placements (on legged robots). Finally, the planned robot pose or joint angles are tracked by a low-level motion controller: model predictive control (MPC) [9] or whole body control methods (WBC) [4] are popular choices for unstable, highly dynamical platforms such as legged robots.

While there have been great results from previous works that adopt the three-stage control pipeline, it often leads to a system that is overly complex and requires significant manual effort to develop. For instance, SLAM algorithms require careful parameter tuning to achieve a balance between latency and accuracy for mobile robots [10]. On the other hand, recent developments in reinforcement learning (RL) based methods open an alternative path towards creating vision-based locomotion controllers without relying on terrain reconstruction or extensive prior knowledge for foothold selection. Researchers have proposed learning-based algorithms that teach robot arms to retrieve objects from cluttered environments [11] and teach drones to avoid obstacles [12] using vision input. They demonstrate the great potential in an end-to-end learning approach to achieve a low-latency control pipeline and reduce the prior knowledge required to design a working controller.

Inspired by this recent progress, we propose a novel control architecture that enables legged robots to successfully solve various visual-locomotion tasks. Specifically, we adopt the philosophy of end-to-end learning and merge the perception and motion planning modules using a neural network (NN). Our contribution is a learnable hierarchical system that contains two individual layers: a high-level vision policy and a low-level motion controller. The high-level vision policy takes two depth images and outputs the desired pose of the robot's base and foothold placements for all swing legs, thereby eliminating the need for a complex SLAM algorithm. The low-level locomotion controller takes the high-level vision policy output, and computes the target motor positions and torques to achieve the desired states. This hierarchical approach allows us to achieve dynamic locomotion in challenging simulated environments including randomly placed stepping stones, staircases, and moving platforms. We also demonstrate zero-shot sim-to-real transfer of visual locomotion policies on the real hardware for walking over stepping stones and climbing up a platform.

## 2   Related Work

Recently, visual-locomotion researchers have developed various promising approaches that tackle the problem using on-board sensors [1, 2, 3, 4, 5, 6, 7, 13, 14, 15, 16]. A main idea in many of these works is to perform explicit terrain shape reconstruction (i.e. local SLAM). For example, Fankhauser and Hutter [13] developed a "Grid Map" stack to construct local elevation maps around the robot and Kim et al. [14] utilized an Intel RealSense D435 depth sensor and a T265 tracking camera to obtain an elevation map of the robot's surroundings. In contrast, we do not employ any explicit mapping mechanism: our learning based visual policy consumes raw depth images from onboard cameras for decision making. Because we eliminate the highly complex mapping process which requires special expertise to tune, our system has a simplified and low latency data pipeline with less chance for manual error accumulation.

Given a representation of the environment provided by a perception module, the controller needs to plan a sequence of leg movements that can guide the robot through the environment. Unlike our approach where perception and motion planning are subsumed into a single neural network, previous works utilize heuristic-based approaches or local optimization to plan adequate motions [5, 3, 16, 2, 7, 17, 18, 19]. For instance, Jenelten et al. developed a metric for scoring each point on an elevation map based on its roughness, closeness to an edge, as well as slope degrees. They then employed an online batch optimization to choose the best landing positions for the feet. In contrast, our learning system directly learns successful foot targets from reward, which greatly simplifies the system architecture. Similar to our approach, Tsounis et al. also proposed to learn a high-level planner module to plan the gait sequence and a low-level gait controller module to control the robot. They demonstrated impressive simulated results of traversing a variety of terrains using a height map as input in simulation. In this work, we focus on demonstrating visual-locomotion skills on a real-robot, which led us to different design choices such as depth images input, an MPC-based low-level controller, and the development of a sim-to-real transfer pipeline.

Foothold optimization is traditionally performed within a single swing step, however, recent works have proposed multi-step contact sequence optimization [3, 16, 21] to account for upcoming changes

in the terrain. For instance, Nguyen et al. proposed a 2-step gait optimization to plan the footstep as well as transition between walking gaits. They demonstrated the approach on a biped robot walking over uneven terrains. In our work, we train the perception and planning module together using deep reinforcement learning, which endows the policy with implicit long-horizon planning capabilities. Although neural networks were also employed in prior works [2, 7] to decide optimal footholds, our system differs in that it also decides the desired body pose and speed. By simultaneous adjusting body pose and footholds, our system can learn challenging visual locomotion tasks such as random stepping stones, which is deemed as difficult to solve with constant body velocities [5],

In addition to methods that leverages the vision input for controlling the robot, recent works have also shown remarkable results for training robots to traverse some uneven terrain without relying on visual perception [22, 23]. By carefully designing the reward function and training procedure, these methods produces policies that can robustly handle moderate uneven terrains by adjusting leg movements. Despite the impressive results demonstrated in these works, vision inputs are still beneficial for efficiently and safely traversing general unstructured terrains. For some tasks, such as walking over gaps, visual guidance is still necessary.

# 3 Methodoloy

## 3.1 Overview

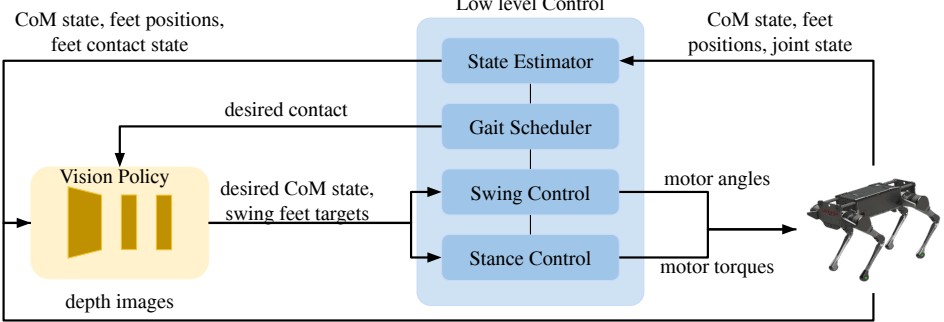

Figure 2: Overview of the visual-locomotion control architecture.

In this work, we adopt a hierarchical architecture for vision-locomotion control, as illustrated in Figure 2. The controller consists of two components: a high-level vision policy and a low-level motion controller. The high-level vision policy uses raw depth images from the onboard cameras and robot state to decide the desired center of mass (CoM) pose (and speed) and landing positions of the swing feet (Section 3.2). Then the low-level motion controller tracks the desired CoM state by combining a position-based swing leg controller and an model predictive control (MPC) based stance leg controller (Section 3.3). The vision policy runs at 20 Hz, while the motion controller runs at 250 Hz. We train our policy entirely in a physics simulation environment and transfer the trained policy to a real robot platform using techniques described in Section 3.4. Unless otherwise specified, all quantities are defined in the yaw aligned inertial frame (Appendix A.1).

## 3.2 Learning-Based Vision Policy

We formulate the visual-locomotion task as a Partially-observable Markov Decision Process (POMDP), $(O, S, A, R, P, p_0, \gamma)$, where $O$ is the observation space, $S$ is the state space, $A$ is the action space, $R : S \times A \mapsto \mathbb{R}$ is the reward function, $P : S \times A \mapsto S$ is the dynamics equation, $p_0$ is the initial state distribution and $\gamma$ is the discount factor. Our goal is to find a policy $\pi : O \mapsto A$ that maximizes the expected accumulated reward: $J(\pi) = \mathbb{E}_{\tau=(\mathbf{s}_0, \mathbf{a}_0, ..., \mathbf{s}_T)} \sum_{t=0}^{T} \gamma^t r(\mathbf{s}_t, \mathbf{a}_t)$.

**Observation and action spaces.** We design the observation space in our experiments as $O = (\mathcal{I}_{1,2}, \mathbf{q}_s, \dot{\mathbf{p}}, \dot{\Phi}, \dot{\Theta}, \mathbf{r}_{1...4}, c_{1...4}, \phi_{1...4}, \mathbf{a}_{prev})$, i.e the two depth images $\mathcal{I}_{1,2}$ from depth sensors shown in Figure 1 (to cover terrains near both front and rear legs), a subset of the CoM pose

$\mathbf{q}_s = (p_z, \Phi, \Theta)$ (CoM height, roll, and pitch), estimated CoM velocity $\dot{\mathbf{p}}$ (Appendix C), gyroscope readings $\dot{\Phi}, \dot{\Theta}$ , the robot's feet positions $\mathbf{r}_{1...4}$, the feet contact states $c_{1...4}$ (one if in contact and zero otherwise), the phase $\phi$ of each leg in its respective gait cycle (Appendix A.2), and the previous action. The action space is $(\mathbf{q}_s^d, \dot{\mathbf{q}}^d, \mathbf{r}_{1...4}^d)$, i.e. the desired CoM pose, velocity, and $i$th foot's target landing position $(r_{xi}, r_{yi})$. The $z$ component of the landing location is inferred from the depth readings: we compute the point cloud from the depth image near the predicted $(x, y)$ coordinate and take the average height of four nearest points. This enables us to achieve better exploration by eliminating invalid foothold targets (e.g. inside an obstacle or high in the air). We find this to be critical for the policy to traverse highly uneven terrains such as stairs.

**Reward function.** We design a reward function:

$$R(\mathbf{s}, \mathbf{a}) = clip(\dot{p}_x, -\dot{p}_x^{max}, \dot{p}_x^{max}) - w_1(|p_y| + |\dot{p}_y|) - w_2|\Psi|, \qquad (1)$$

where $\dot{p}_x$ is the CoM velocity in the forward direction, $p_y$ the CoM displacement in the lateral direction, and $\Psi$ the base yaw angle. The first term rewards the robot to move forward with a maximum speed controlled by $\dot{p}_x^{max}$, the second term penalizes the robot from moving sideways, and the last term encourages the robot to walk straightly; $w_1, w_2$ modulates the importance of different terms. In our experiments, we chose $\dot{p}_x^{max} = 0.375$ m/s, $w_1 = 1.25$, and $w_2 = 0.125$.

**Early termination.** A training episode is terminated if: 1) the robot loses balance (CoM height $p_z$ below 0.15 m, pitch $|\Theta| > 1$ rad, or roll $|\Phi| > 0.3$ rad in our experiments), 2) the robot steps within 0.02 m of the boundary of the stepping stones or stairs, or 3) the robot reaches an invalid joint configuration, e.g. knee bending backwards.

### 3.3 Motion Controller

The motion controller computes appropriate motor position and torque commands to achieve the desired landing positions of the swing legs and the target base poses from high level policy. Our motion controller separately controls swing and stance legs. The swing leg controller computes the feet positions by interpolating a time based curve $\alpha(t)$ between the swing start and target landing positions. Instantaneous feet positions are converted to desired joint angles through inverse kinematics (IK) and tracked using proportional-derivative (PD) control. The stance leg controller, on the other hand, achieves the desired position and velocity of the robot base by computing sequences of contact forces between the feet and the ground. By approximating the robot dynamics using Centroidal Dynamics Model (CDM), we formulate this problem as a convex model predictive control (MPC) problem similar to [24]. The optimized contact forces are mapped to stance leg joint torques using Jacobian Transpose. More details regarding the motion controller can be found in Appendix A.

### 3.4 Sim-to-real Transfer

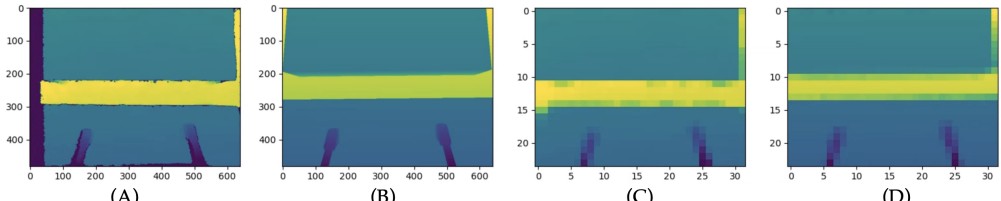

Figure 3: A post-processing technique to reduce the sim-to-real gap in perception. (A) An original depth map obtained from the D435 camera with an invalid band on the left of the image and noises near edges. (B) A simulated depth map. (C) The filtered and down-sampled D435 depth image. (D) The filtered and down-sampled simulation depth image.

Sim-to-real transfer is a persisting problem in robotics, and is especially hard for visual locomotion problems because of distribution shift in image space and discrepancies in robot dynamics.

Images from simulated and real-world sensors can be significantly different (Figure 3, A and B). To bridge the gap, we adopt a post-processing procedure that maps both simulated and real depth images to a similar domain. As shown in Figure 3, C and D, we first add random Gaussian noise

and randomly paint pixels black in the simulated depth image. To mimic the noise around the object edges in a stereo depth camera, pixels along an edges in the depth image identified by a Canny edge detector have a higher probability of being dropped. We then apply an in-painting operation for both simulated and real images to fill the missing pixels[25], followed by a down-sampling operation. This transforms the depth images in simulation and real world to a similar distribution.

Discrepancies in dynamics such as mismatch in friction make precise foot-placements difficult. For example, we found the average difference of feet landing positions in the world frame can be $2 \sim 4$ cm between sim and real when the same sequence of actions are executed. This is enough to trigger a failure in the step-stone task. To compensate this issue, we adopt the domain randomization technique [26] to train a robust policy with randomized simulation parameters. More details regarding the randomization process can be found in Appendix D. Adding randomization not only allows the policy to be more robust, but also enables the vision policy to observe more diverse states.

## 4 Experiment and Results

### 4.1 Experiment Setup

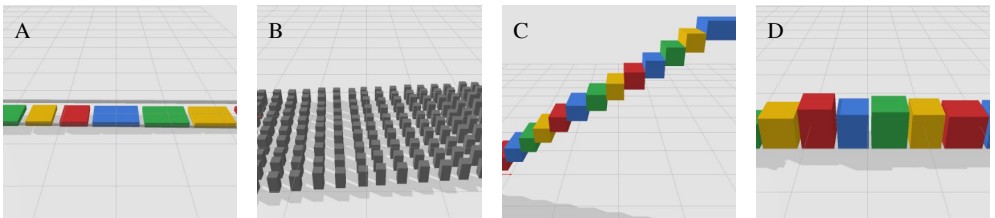

Figure 4: Examples of environments used for visual locomotion tasks: randomly placed stepping stones, quincuncial piles, staircases, and uneven terrains.

We evaluate our method on the Unitree Laikago [27] quadruped robot, which weighs $24kg$. The robot is equipped with 12 actuated joints. To collect visual data, we install two depth cameras on the robot: one Intel D435 in the front for a wider field of view and one Intel L515 on the belly with better depth quality in close range (Figure 1). At inference time, we process all depth images as described in Section 3.4, which results in two $32 \times 24$ depth images as inputs to the policy.

We train our hierarchical policies using the PyBullet physics simulator [28] with a distributed implementation of augmented random search (ARS) [29]. We use $N = 256$ perturbations per ARS iteration and run the algorithm until convergence with a maximum of 2000 training iterations, amounting to $1,024,000$ simulation episodes per trial. We use a Multi-layer Perceptron (MLP)-based policy architecture with $54,656$ parameters. For all experiments, we run ARS with a grid search for four hyper-parameters, resulting in 12 trials. We report the performance of top-3 policies by testing them on 150 randomized environments for each task. More training details can be found in Appendix E.

For experiments in both simulation and real world, we train policies with perception noise. We then fine-tune them with the dynamics randomization scheme as described in Section 3.4 before deploying to the real robot. We empirically find this helpful for faster training convergence. The fine-tuned policy is deployed on the robot without the need of additional hardware data.[1]

### 4.2 Simulation Results

We first evaluate the capability of our learning system on a variety of challenging visual locomotion tasks, including walking over randomly placed stepping stones, quincuncial piles, uneven terrains, and moving platforms. A subset of the simulation environments are shown in Figure 4. We train separate policies for each environment and show the statistical results in Figure 5. To measure the performance of each policy, we design a metric, *performance ratio*, as $\frac{p_x^T}{p_x^{max}}$, where $p_x^T$ is the distance travelled by the policy in the environment, and $p_x^{max}$ is the maximum distance the policy could reach, e.g. where the terrain ends.

---

[1]Videos of our trained policies in simulation and real-world can be found at: https://youtu.be/1X-NH-EuynQ

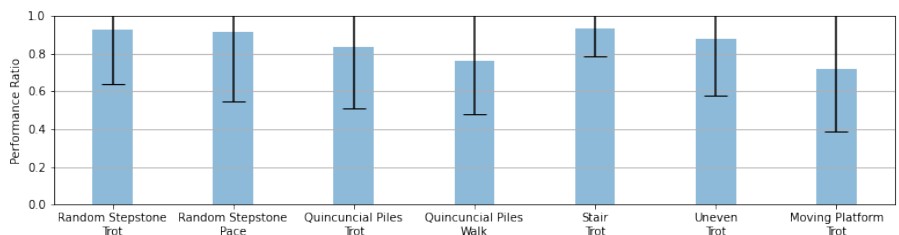

Figure 5: Performance ratio of our policy for different simulated environments.

**Randomly placed stepping stones.** The first task is to walk over a series of randomly placed stepping stones, Figure 4A. The widths, lengths and gap sizes of stepping stones are sampled from $[0.55, 0.7]$, $[0.5, 0.8]$, and $[0.07, 0.2]$ meters respectively. The agent fails the task if the robot steps outside of the stone or into the gaps. Since the stones and their positions are randomly sampled, this task cannot be accomplished without vision. To successfully achieve this task, the robot needs to identify the position and size of the stones and also plan for future footholds. We demonstrate that our method can obtain policies that solves this task using both trotting and pacing gaits.

**Quincuncial piles.** The random stepping stone task evaluates the robot's ability to identify and handle terrain changes in the forward direction. A natural extension is to also include the lateral direction. In the second task we create a grid of 2 dimensional stepping stones (Figure 4B). Each stone has an area of $0.15 \times 0.15$ m$^2$ with a standard deviation of $0.015$ m in height, and is separated by $[0.13, 0.17]$ m from each other in both $x$ and $y$ directions. At the beginning of each episode, we also randomly rotate entire stone grid in $[-0.1, 0.1]$ rad. Despite being significantly more difficult, using our framework we can obtain policies that can traverse the field using trotting or walking gaits.

**Uneven terrains.** So far the tasks focus on evaluating the robot's ability to avoid undesired regions on relatively even terrains. To test the policy's performance in handling different heights, we designed a staircase climbing task (Figure 4C). The depth of each stair is uniformly sampled in $[0.25, 0.33]$ m and the height in $[0.16, 0.19]$ m to mimic the dimensions of real-world stairs. This task is quite challenging for Laikago robot because each stair is as tall as the robot's knee joint. Figure 4D shows another environment we designed for evaluating the ability of our approach to handle uneven terrains. In this environment, the height offsets of neighboring stones are uniformly sampled in $[-0.13, 0.2]$ m, and a gap of $[0.05, 0.1]$ m is added between the stones.

**Moving platforms.** One benefit of using vision input to the policy is that it can potentially handle moving objects better than methods that relies on explicit terrain reconstruction. To demonstrate the capability of our learning system, we take the random stepping stone environment (Figure 4A) and allow each piece to move. Each platform follows a periodic movement whose magnitude and frequency are randomly sampled in $[0.10, 0.15]$ m and $[0.4, 1.0]$ Hz, respectively. Also, we randomly pick half of the platforms to move horizontally and the rest vertically. This task requires the robot to infer both the position and velocity of the platforms. To facilitate learning, we stack a history of three recent images as input to the policy for this task. As shown in the accompanying video, our policy learns to identify the moving objects in the scene and will slow-down and wait for the platforms to reach an ideal location before striding.

### 4.3 Comparison to baseline methods

We compare our proposed method to two baselines: end-to-end training and heuristics-based foot placement. For the sake of simplicity yet without loss of generality, all comparisons are done in the uneven terrain environment (Figure 6 Top), which captures the difficulty of both the stepping stones and staircases. The results can be found in in Figure 7.

#### 4.3.1 End-to-end training

In the first baseline, we train an end-to-end neural network policy that takes the images and the robot states as input (same as our hierarchical policy) and outputs desired motor angles. We choose the architecture of Policy Modulating Trajectory Generator (PMTG) [30], which has demonstrated

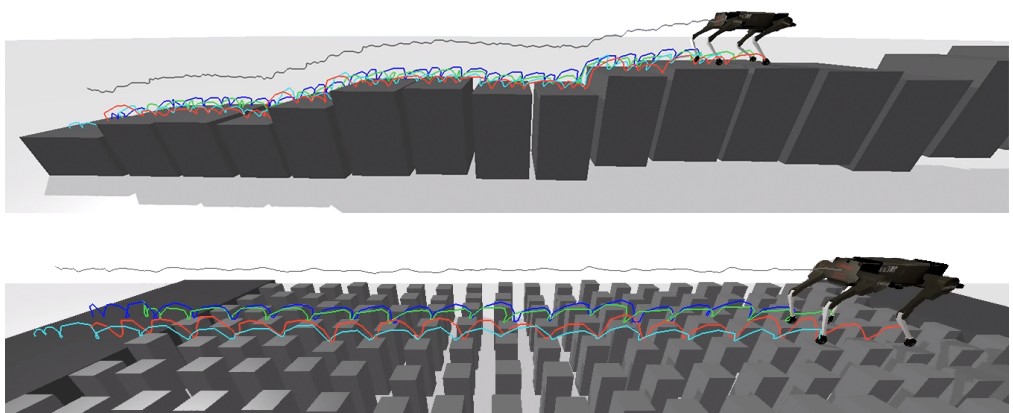

Figure 6: Laikago robot walking over uneven terrain and randomized pillar terrains. The black curve refers to the CoM trajectory, and colored curves represent the feet trajectories.

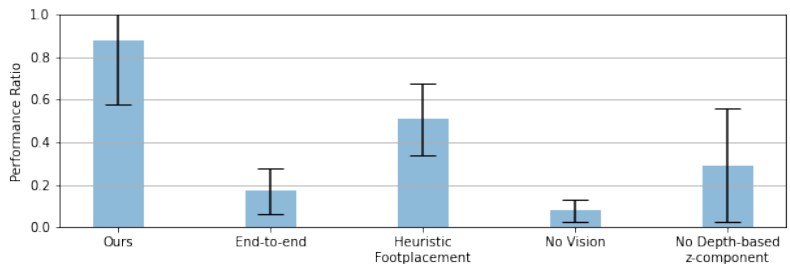

Figure 7: Comparison of performance between our method and the baseline methods.

high-quality and transferrable locomotion policies [22]. Since PMTG generates the trajectory in the joint space, it was not able to achieve precise foot placement that is needed for many of our testing environments, as shown in Figure 7.

### 4.3.2 Heuristics-based foot placement

In the second baseline, we compare our method to prior approaches that adjust the foot placement by computing a score map of surrounding terrains [2, 4, 18]. In our baseline implementation, we define a point on the nearby terrain to be valid if 1) it is within reach of the swing leg, 2) the z-component of the surface normal is larger than 0.9 (i.e. pointing upward), and 3) the standard deviation of the nearby heights (i.e. roughness) is less than 0.05m. For better comparison, we use the same motion controller as in our approach and train a high-level policy to output the desired base pose and velocity using the same observation space and learning procedure. As shown in Figure 7, using heuristics alone to determine footplacement got lower score than our proposed approach. One important reason is that this heuristics does not take the base movement of the robot into consideration. As a result, it may propose landing positions that are incompatible with the robot's CoM speed.

### 4.4 Ablation Study

A key hypothesis we make in this work is that, despite that the images have low resolution, they contain critical information that enables our policy to traverse a large variety of environments. To validate this hypothesis, we first perform an ablation run by removing the vision component from the observation space and retraining a vision-less policy. As seen in Figure 7, without vision input, the policy is not able to accomplish the task, indicating the importance of having vision as input.

Another key component in our method is to use the depth input for inferring the z-component of the foothold location. This creates a more meaningful action space for the policy to explore during learning. As shown in Figure 7, the performance drops significantly we do not use depth to infer the z-component of the foot placement.

### 4.5 Validation on Real Robot

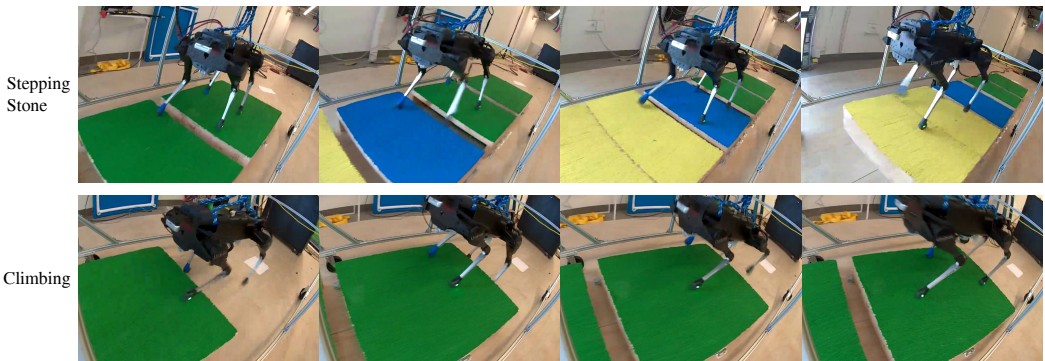

Figure 8: Laikago robot solving two challenging visual-locomotion tasks.

We deploy the trained policy to a Laikago robot for two tasks, walking over gaps and climbing onto stepping stones (see supplementary video). Our real-world setup (Figure 8 Top) consists of four stepping stones that are separated by three gaps with widths between $[0.12, 0.18]$ m.

In the first task, the performance is measured by the number of gaps that are crossed successfully, summed over legs. For instance, if all four legs cross one gap and the robot is completely on the next stepping stone, the score would be 4. Therefore, the upper bound of the score is 12: three gaps times four legs. An experiment episode is terminated if the robot falls, any leg steps into the gaps or the robot steps outside the stones. As a baseline, the score is $0.9 \pm 1.4$ for a blind-locomotion policy: it can rarely clear a single gap. Over eight real-world episodes, our policy is able to consistently achieve a score of $\mathbf{10.1 \pm 2.2}$. There are two episodes in which the robot completes the full course with a score of 12; For seven out of eight episodes, the robot reaches the last stepping stone. The real-world failure cases are mainly caused by the robot stepping too close to the edge of the terrain and slipping into the gap. This is likely due to the discrepancies in the dynamics and vision model. Despite applying domain randomization techniques to mitigate the model discrepancy, we found precisely controlling the landing position of the feet for a moving robot still challenging.

In the second task, the robot needs to climb onto a stepping stone, which is 0.18 m above from the ground (Figure 8 Bottom). Similarly, we also define the performance score as the number of legs reached the top of the stone (4 is the max). The learned policy reaches a score of $\mathbf{3.7 \pm 0.7}$ over seven experiment episodes. The robot successfully steps onto the stone for six out of the seven times. In the only failure case, the robot loses balance because its rear foot hits the edge of the stone.

## 5 Conclusion and Future Work

In this work, we present a hierarchical learning system to tackle challenging visual-locomotion tasks. By using a high-level learned vision policy that consumes raw camera images, we eliminate the need to explicitly construct 3D terrain maps, and thus reduce the control latency and architecture complexities, and more importantly, can handle dynamically moving terrains. The low-level, model predictive control based motion controller greatly reduces the motion tracking error on the hardware and thus narrows the sim-to-real gap. We demonstrate that policies trained using our system can reliably walk over challenging terrains such as stepping stones that require fine visual-motor control.

One limitation of this work is that the gaits are manually specified and fixed for each task. In contrast, animals can dynamically modulate their walking pattern to adapt to changes in the terrain. In the future, we plan to extend our framework to enable the visual policy to adjust the gait pattern online. Recent works have shown great potential in representation learning for vision-based robotic control problems [31, 32]. For example, Hoeller et al. trained a VAE to encode the environment for a legged robot navigation task. Incorporating these techniques can further improve the learning performance of our framework. Additionally, animals can traverse complex terrains while paying attention to a small area, and can reason about their hind limbs that they cannot see. This makes adding memory [33] and attention [34] to the policy architecture another promising future direction.

## Acknowledgement

We would like to thank Gus Kouretas, Thinh Nguyen, Noah Brown, Satoshi Kataoka, and the Operations team at Robotics at Google for the help in setting up the testing environment, debugging robot hardware and camera issues. We would also like to thank Ken Caluwaerts, Krzysztof Choromanski, Kuang-Huei Lee, Daniel Ho, Yuxiang Yang, and the anonymous reviewers for valuable discussion and suggestions.

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
