# OpenReview forum: "Visual-Locomotion: Learning to Walk on Complex Terrains with Vision"
_robot-learning.org/CoRL/2021/Conference — CoRL2021 Poster_

### Official Review · Reviewer_2GnU · 2021-07-22

**Originality:** Good
**Technical Quality:** Very Good
**Clarity Of Presentation:** Good
**Impact:** 4

**Recommendation:**

Strong Accept: I recommend accepting the paper and will argue for my recommendation even if other reviewers hold a different opinion.

**Summary:**

This paper proposes a hierarchical vision-based controller for quadruped locomotion on a variety of random terrains. The high-level component is trained using DRL and specifies the walking pattern for the legs and the robot's base. The low-level component separately controls the swing legs and the stance legs to realize the movement asked by the high-level controller. The system is successfully tested on a robot using zero-shot sim2real transfer.

**Issues:**

- Explain the training process details in the main paper body.
- The policy archicture is only briefly explained in the appendix without any accompanying text. For example, I'd like to know if the vision inputs are processed using CNN networks.
- If possible, test the approach with other SOTA RL approaches such as SAC & TD3.

**Reviewer Expertise:**

Very good: Comprehensive knowledge of the area

**Strengths And Weaknesses:**

Strengths:
- The system design is straightforward.
- The observation and action spaces are defined in a compact way.
- The approach is tested in a wide range of randomized environments with good results. Comparision to the baselines and the ablation study look convincing as well.
- The paper is well-written. I couldn't find any typos.

Weaknesses:
- The training details are only explained in the appendix. Even there the paper does not report any learning curves or the number of simulation budget for the training.
- It seems that the approach only supports the gait that is encoded in the gait scheduler.
- It would have been nice to see how the model works using other SOTA RL algorithms such as SAC & TD3.

**Summary Of Recommendation:**

I liked the paper since it is well-written, the approach is clear, and the results are looking good. I am a bit suspecious that the learning contribution is minor, but that should be OK cosidering the challenges of deploying such a system on an actual robot. I am happy with the paper as it is, but below you can find my suggestions for improving it.

---

> ### Author Response · Authors · 2021-08-30
> **Thank you for your review!**
>
> Thank you for the great suggestions and valuable comments that help us improve the quality of our work! Please find below a list of issues mentioned in the review and our response to them.
>
> Q: **Explain the training process details in the main paper body.**
> A: We have revised the paper to include more training details in the main paper body.
>
> Q: **The policy architecture is only briefly explained in the appendix without any accompanying text. For example, I'd like to know if the vision inputs are processed using CNN networks.**
>
> A: Our vision inputs are processed using an MLP encoder. We have also experimented with other architectures such as CNN and found them perform similarly in the simulation tasks we designed. We have included more details and discussions in the corresponding appendix section.
>
> Q: **It seems that the approach only supports the gait that is encoded in the gait scheduler.**
>
> A: The reviewer is correct that our method currently is limited to fixed gait schedulers and allowing more flexible locomotion gait enable the robot to traverse a more diverse set of environments, which is an important future direction for our project.
>
> Q: **It would have been nice to see how the model works using other SOTA RL algorithms such as SAC & TD3.**
>
> A: Thank you for the suggestion! We have tried applying SAC to our problem during the experiments but did not achieve satisfying results. This is mainly because we use a Partially-Observable MDP (POMDP) for our problem formulation, which led to unstable training for SAC in our experiments. As such we chose ARS to generate results in this work, which can better handle non-markovian cases.

---

> > ### Comment · Reviewer_2GnU · 2021-09-03
> > **Happy with the updates**
> >
> > Thanks for your reply, and for taking the reviewers' suggestions into account. I change my vote to strong accept.

---

### Official Review · Reviewer_MZ8r · 2021-07-23

**Originality:** Very Good
**Technical Quality:** Very Good
**Clarity Of Presentation:** Very Good
**Impact:** 3

**Recommendation:**

Weak Accept: I recommend accepting the paper, but will not argue for my recommendation if the majority of other reviewers have a different opinion.

**Summary:**

The paper provides a hierarchical control that combines a high-level vision-based planner and a low-level MPC controller. The planner is an RL policy that takes the processed raw image data to decide the target CoM pose trajectory and the target feet locations. The main contributions include:
1. Use raw image data as the policy input instead of a processed height map.
2. Learn not only the foot location but also the target CoM trajectory.

**Issues:**

1. Please discuss citation [18]. Even if it uses heuristic foot placement, but performance-wise, it looks no different than the current method.
2. Please elaborate more about the action space. The way it is present in line 115 looks like a vector with dimension 20, but MPC requires a sequence of target states as input.
3. Sec 2 second paragraph, should you cite DeepGait?
4. Line 112, there is no Appendix A.5. Should be A.4
5. Appendix line 444, why it is SQP, not a QP? My understanding is SQP is a nonconvex problem, but the formulation looks the same as MIT chhetah3 convex-MPC formulation.

**Reviewer Expertise:**

Excellent: Expert knowledge on the topic of the paper

**Strengths And Weaknesses:**

The paper is written clearly with enough discussion, explanation. I do appreciate the detailed appendix that provides useful information to elaborate the content in the main paper. I think the control method described in the paper is reasonable and the simulation results are good. I support the argument in the paper that learn both CoM pose and foot location is better than learn foot location only.

For the weak points, the paper may miss one case in the discussion. If we two actions: CoM pose and foot location and two controllers: learned and heuristic, we can come up with four combinations: 1. learned both CoM pose and foot location; 2. learned CoM pose, heuristic foot location; 3. heuristic CoM pose, learned foot location; 4. heuristic both CoM pose and foot location. The paper's method is 1. and the baselines are 4 and most of the literature is 3. However, citation [18] is case 2 where learn only and CoM pose and can pass all the tasks in the current papers. I would suggest discussing [18] in 4.3.2 Heuristic-based foot placement. It would support the last sentence that landing positions should compatible with the robot's CoM speed.

Other weakness points are:
1. Prefer more hardware experiments, like stair climbing. The current experiments are easy. The simple QP stand control + Raibert's heuristic foot placement can complete the tasks. The gap avoidance just needs to find the nearest feasible ground to adjust the target landing positions. I have tested this on A1.
2. Prefer elaborating more about the action space. The MPC requires a trajectory of CoM and foot locations, not just an instant one. The action space in line 115 looks like a single state of CoM and foot locations. So just to be clear, does the policy output q_d(t=1) or {q_d(t=1), q_d(t=2), ...q_d(t=H))? So does the foot location.

**Summary Of Recommendation:**

The paper provides a clean hierarchical control strategy to include raw images in the legged locomotion control. Learning both CoM and foot location is useful.

---

> ### Author Response · Authors · 2021-08-30
> **Thank you for your review!**
>
> We want to thank the reviewer for the insightful feedback and detailed comments! We have provided below a response to the issues listed in the review and hope they can help address your concerns!
>
> Q: **More challenging hardware experiments.**
>
> A: Thanks for the suggestion! In this work, we focus on two representative tasks for visual-locomotion and found that due to the sim-to-real gap and noises in the real-world environments and sensors, it is a non-trivial task to achieve perfect performance for these tasks. We agree that further improving and demonstrating our method in more challenging and diverse tasks in the real world is an important future work direction.
>
>
> Q: **Comparison to heuristics-based foot placement baseline with learned CoM target [1].**
>
> A: Thank you for the nice summary of different combinations for controlling CoM pose and foot placement! The baseline in our work does learn a policy to predict the CoM target and use a heuristics-based approach for adjusting the foot placement. We recognize that [1] used a similar approach and achieved impressive results in a variety of environments. One factor that potentially contributed to the performance gap in our experiment is that we used the noisy simulated depth images to adjust the foot placement, which may lead to imprecision in the landing location.
>
>
> Q: **Please elaborate more about the action space. The way it is present in line 115 looks like a vector with dimension 20, but MPC requires a sequence of target states as input.**
>
> A: The high-level vision policy outputs a desired future robot state (0.05s from current) and MPC tracks the target by creating a reference trajectory with the same target state at every timestep. We found this scheme to achieve the best tracking performance. We have also experimented with interpolating the reference trajectory from the current state, yet it achieved worse tracking results.
>
>
> Q: **Include DeepGait [2] in literature review.**
>
> A: Thank you for noting the missing related work! [2] is indeed quite relevant to our work and we have included it in our literature review. [2] also proposed to decouple the policy into a high-level planner and a low-level gait controller.  Their approach demonstrated impressive simulated results of traversing a large variety of terrains using a height map as input. In this work, we focus on demonstrating visual-locomotion skills on a real-robot, which led us to different design choices such as depth images input, an MPC-based low-level controller, and the development of a sim-to-real transfer pipeline.
>
>
> Q: **Line 112, there is no Appendix A.5. Should be A.4**
>
> A: Thanks for spotting this! We have corrected it in the revision.
>
> Q: **Appendix line 444, why it is SQP, not a QP? My understanding is SQP is a nonconvex problem, but the formulation looks the same as MIT chhetah3 convex-MPC formulation.**
>
> A: Our formulation is indeed solving a QP instead of SQP. We have corrected this in the revised text.
>
>
> [1] Glide: Generalizable quadrupedal locomotion in diverse environments with a centroidal model. Xie et al. 2021.
> [2] DeepGait: Planning and Control of Quadrupedal Gaits using Deep Reinforcement Learning. RA-Letters 2020.

---

> > ### Comment · Reviewer_MZ8r · 2021-09-03
> > **Final recommendation**
> >
> > Thanks for the update. The feedback resolves my questions.

---

### Official Review · Reviewer_jPiH · 2021-07-24

**Originality:** Good
**Technical Quality:** Good
**Clarity Of Presentation:** Good
**Impact:** 3

**Recommendation:**

Weak Accept: I recommend accepting the paper, but will not argue for my recommendation if the majority of other reviewers have a different opinion.

**Summary:**

This paper presents a reinforcement learning based **visual policy** for quadrupedal robots to walk on **structured rough terrain** with discrete footholds such as stepping stones and stairs with varying step lengths and step heights. The learning based visual policy takes in two depth images (one each attached near the front and rear of the robot) and provides high-level commands such as desired center-of-mass positions and velocities and swing foot positions to a low-level MPC controller. The policy is validated in a wide range of scenarios, both, in simulation and hardware on the Unitree Laikago quadruped and compared against a heuristic policy and an end-to-end trained policy.

**Issues:**

It would be good to address the comments under the listed 'Weakness' of the paper.

**Reviewer Expertise:**

Excellent: Expert knowledge on the topic of the paper

**Strengths And Weaknesses:**

The primary strength of this paper is that it does a good job of experimentally validating their policies on hardware, and also comparing against relevant baselines.

As also suggested in the paper, the main idea behind using a learning based approach is the avoidance of building an explicit elevation map and planning footsteps/base motions using that map. Instead, the visual policy outputs the plan for the footholds and base motions directly from the depth camera images and robot states. However, this is not very new and several works have previously looked at combining high-level visual inputs to low-level model based controllers like MPC.

I see the following limitations:

1. Related Works: There are key references missing from the related works and which might help the authors to further improve on their research as well. See end of this section for the list of additional references and below for a summary:

  (a) Learning based approaches:

  [R1] DeepGait - This paper introduces a learning based approach for legged robots to navigate on discrete terrain using the local height map data as the visual input and outputs desired base positions, velocities, foot positions and gait timing variable, which are then fed to a low-level controller. I believe this is very relevant and similar to the proposed method (except for the visual input representation).

  [R2] This paper presents a learning based approach for visual navigation in cluttered environment that uses a VAE to build a predictive model of the depth images. This is then used to train a model-based RL policy that outputs high-level commands for a low-level model-based controller.

[R3] is also very relevant to this current manuscript, where a highlevel CNN policy is used along with a low-level MPC controller.

  (b) Control theoretic approaches [R4]-[R5]: There's quite a few works on using control theoretic (and with non-visual inputs) for legged robots to walk on discrete terrain like stepping stones. I believe the authors can use some ideas from these works to improve on the foot tracking performance without significant domain randomization of the dynamics parameters.

2. Dynamic capabilities of the locomotion controller:

The learned policies only consider motions along a straight line. Are there any limitations on the turning capabilities of the policy (especially on the terrain with the stepping stones)? The authors should consider stating the limitations of the locomotion controller.

3. MPC as a limitation to learn more complex behaviors: As stated in the manuscript, the current policy only considers gaits of a particular contact schedule (trotting/static walk/etc). In this regard, it seems like the MPC places additional constraints and limitations on the achievable robot behaviors. It would be interesting to see if this limitation can be lifted if desired joint trajectories can be learned from visual inputs. The authors do consider an end-to-end approach, however, it is unclear why this has poor performance.

4. Details on the policy implementation:
It would be good to have some more info on whether the current policy can be run on-board or requires compute on an off-board workstation.

5. CoM x-y position estimation:
It would be good to have some details on how the x-y positions are estimated on the robot since this is one of the inputs to the vision policy.

6. Requirement of 2 cameras: Although this is a good engineering solution, I believe that this simplifies the learning problem quite a bit.


[R1] Tsounis, Vassilios, et al. "Deepgait: Planning and control of quadrupedal gaits using deep reinforcement learning." IEEE Robotics and Automation Letters 5.2 (2020): 3699-3706.

[R2] Hoeller, David, et al. "Learning a state representation and navigation in cluttered and dynamic environments." IEEE Robotics and Automation Letters 6.3 (2021): 5081-5088.

[R3] Villarreal, Octavio, et al. "MPC-based controller with terrain insight for dynamic legged locomotion." 2020 IEEE International Conference on Robotics and Automation (ICRA). IEEE, 2020.

[R4] Grandia, Ruben, et al. "Multi-layered safety for legged robots via control barrier functions and model predictive control." arXiv preprint arXiv:2011.00032 (2020).

[R5] Nguyen, Quan, et al. "Dynamic Walking on Randomly-Varying Discrete Terrain with One-step Preview." Robotics: Science and Systems. Vol. 2. No. 3. 2017.


**Summary Of Recommendation:**

While the paper presents some good experimental results, there are several works that look into visual locomotion for legged robots in challenging terrain with discrete footholds. The use of two cameras simplifies the learning problem and avoids having to create an explicit map or use RNNs/LSTMs.

---

> ### Author Response · Authors · 2021-08-30
> **Thank you for your review!**
>
> We really appreciate your time and your valuable feedback that helps us improve our work! We have listed the main concerns raised in the review and provided our response below:
>
> Q: **Additional related works**
>
> A: Thank you for suggesting the additional references! They are indeed very interesting and relevant work and we especially appreciate your time in providing the summaries! We have incorporated them into the related work and conclusions of the revised manuscript.
>
> Q: **Dynamic capabilities of the locomotion controller (for example turning):**
>
> A: Thank you for the comment! Our locomotion controller does handle turning by specifying a desired yaw rotation speed of the robot’s base. To demonstrate this, we have trained a policy for a curvy quincuncial environment, where the high-level vision policy learns to turn the heading of the robot to follow the path. We have included a clip of the result in the updated supplementary video.
>
> Q: **Limitations introduced by using MPC; why end-to-end doesn’t train well?:**
>
> As mentioned by the reviewer, one limitation of our approach is that the available robot behavior is constrained by the expressiveness of the low-level locomotion controller. Further improving our algorithm to enable a diverse set of robot behaviors is an important future direction we plan to pursue. Learning to map from vision input to motor commands would provide more flexibility in robot behavior, however, we found that directly training such an end-to-end policy is challenging in practice. This is likely because the more flexible action space introduces a much harder exploration problem.
>
> Q: **Can the current policy be run on-board or require compute on an off-board workstation?**
>
> A: Our policy is executed on a Zotac Magnus EN2070 mobile PC, which is 0.2x0.06x0.21m in size and weighs around 3kg. During our experiments, we placed the PC on the safety rack. However, it is possible to mount the PC on the robot as the weight and size is well within the load capacity of Laikago according to Unitree Spec (>= 9kg). We have included this in the updated appendix.
>
> Q: **How is the CoM x-y position estimated?**
>
> A: The CoM-related part of our observation space include: CoM height (estimated from local foot positions), estimated CoM velocity, roll and pitch of the robot base, and angular velocity of the robot base from IMU sensor. Therefore, our policy does not take CoM x-y positions as input. We have revised the paper to make this clear.
>
> Q: **Requirement of 2 cameras: Although this is a good engineering solution, I believe that this simplifies the learning problem quite a bit.**
>
> A: Thank you for your feedback! We agree that a single camera solution would lead to a more concise and elegant system. In fact, we have experimented with using a single front camera and representing the vision policy with an LSTM network. We found that training this policy was not significantly more difficult than a two-camera setting. This is potentially due to that we adopted a gradient-free algorithm for policy learning, which avoids the challenges in gradient computation for gradient-based methods. However, the trained LSTM policy needs to remember and infer appropriate foot locations for the rear legs from past observations, which makes it more sensitive to the sim-to-real gap. As a result, we observed poorer real-world performance with this setting.

---

> > ### Comment · Reviewer_jPiH · 2021-09-03
> > **Thanks for the updates; retain original decision**
> >
> > Thanks for addressing the comments; the current version reads well. I would retain my original decision of 'Weak Accept'. While the work seems promising, rigorous experiments on more complex terrains are required on hardware to evaluate and validate the proposed approach.

---

### Official Review · Reviewer_tqbD · 2021-07-24

**Originality:** Good
**Technical Quality:** Good
**Clarity Of Presentation:** Very Good
**Impact:** 3

**Recommendation:**

Weak Reject: I recommend rejecting the paper, but will not argue for my recommendation if the majority of other reviewers have a different opinion.

**Summary:**

Summary: This paper proposes a hierarchical approach to enable visual locomotion from onboard cameras. The high-level planner predicts the desired base pose, velocity and the target for the swing leg at a low frequency which is executed by an MPC running at a high frequency and uses a simplified robot dynamics model. The results are demonstrated in a wide variety of simulated terrains and also on real hardware for the stepping stone tasks.



**Issues:**

As mentioned earlier, I would like to see a discussion on the following points:
1. One limitation of the proposed method is that there is a separate policy trained for each of these seemingly similar tasks. It would be good to know the benefits of the proposed method compared to [14], which uses the same heuristic across their experiments and only uses onboard cameras as well.
2. The paper should at least discuss (and ideally compare) to [A] below which uses the same decoupling of a high-level planner and a low-level controller.
3. It is quite interesting to see a simple visual processing pipeline that is quite effective in bridging the sim to real gap. It would be interesting to understand the limitations of this.
4. The paper should talk about the major failure cases and the limitations of their overall method since the reported accuracies even in the simulation are not 100%.

[A] Tsounis, Vassilios, et al. "Deepgait: Planning and control of quadrupedal gaits using deep reinforcement learning." IEEE Robotics and Automation Letters 5.2 (2020): 3699-3706.


**Reviewer Expertise:**

Very good: Comprehensive knowledge of the area

**Strengths And Weaknesses:**

Comments:
1. The idea of decoupling foot-step planner and leg controller is quite natural and elegant, and seems quite promising from the results presented in the paper.
2. The heuristic controller used in Section 4.3.2 is not very clear and the failure mode analysis of not very convincing. Instead, the authors should use the heuristic method of foot placement as proposed in Section IV-B of [14], which takes the com speed into account before deciding the foot placement.
3. One limitation of the proposed method is that there is a separate policy trained for each of these seemingly similar tasks. It would be good to know the benefits of the proposed method compared to [14], which uses the same heuristic across their experiments and only uses onboard cameras as well.
4. The paper should at least discuss (and ideally compare) to [A] below which uses the same decoupling of a high-level planner and a low-level controller.
5. It is quite interesting to see a simple visual processing pipeline that is quite effective in bridging the sim to real gap. It would be interesting to understand the limitations of this.
6. The paper should talk about the major failure cases and the limitations of their overall method since the reported accuracies even in the simulation are not 100%.
7. The paper is very well written and easy to follow and understand. The figures are great.

[A] Tsounis, Vassilios, et al. "Deepgait: Planning and control of quadrupedal gaits using deep reinforcement learning." IEEE Robotics and Automation Letters 5.2 (2020): 3699-3706.

**Summary Of Recommendation:**

The paper executes the idea high-level learned visual planner coupled with a low-level MPC-based controller quite well and my current inclination is to accept the paper if the heuristic baseline controller is modified as suggested above.

---

> ### Author Response · Authors · 2021-08-30
> **Thank you for your review!**
>
> We thank the reviewer for the detailed comments and insightful suggestions! Please see below our response to the issues raised by the reviewer.
>
> Q: **One limitation of the proposed method is that there is a separate policy trained for each of these seemingly similar tasks. [1] uses the same heuristics across different tasks and only uses onboard cameras as well. It would be good to know the benefits of the proposed method compared to [1].**
>
> A: In this project, we focus on validating our proposed approach for individual visual-locomotion tasks. We agree with the reviewer that a single policy for multiple tasks is important and is indeed what we aim to achieve in future work. The main difference between our approach and [1] is that we use a vision-based policy to determine the foot-step instead of explicitly building a local height map with heuristics-based foot-step selection. By using an RL-based approach to consume vision input, it allows us to better handle dynamic scenarios such as the moving platforms shown in our experiments. In addition, we demonstrate that learning the foot-placement can achieve better performance than a heuristics-based approach for the problem we designed. However, we do recognize that using RL to train such a policy can be time-consuming as the policy needs to explore different foot-placements including ones that are invalid.
>
>
> Q: **The paper should at least discuss (and ideally compare) to [2] which uses the same decoupling of a high-level planner and a low-level controller.**
>
> A: Thank you for bringing up this work! [2] is indeed quite relevant to our work and we have included it in our literature review. [2] also proposed to decouple the policy into a high-level planner and a low-level gait controller. Their approach demonstrated impressive simulated results of traversing a large variety of terrains using a height map as input. In this work, we focus on demonstrating visual-locomotion skills on a real-robot, which led us to different design choices such as depth images input, an MPC-based low-level controller, and the development of a sim-to-real transfer pipeline.
>
>
> Q: **It is quite interesting to see a simple visual processing pipeline that is quite effective in bridging the sim to real gap. It would be interesting to understand the limitations of this.**
>
> A: Thank you for your comment! A main limitation of our current visual processing pipeline is that it is designed mainly for depth-based sensors in indoor environments. As such, for tasks that require additional features such as color images, additional sim2real techniques would be needed. Furthermore, for outdoor applications, the noise patterns seen by depth sensors can be significantly different due to lighting conditions. A different set of sensors and processing pipelines would be needed in these situations.
>
>
> Q: **The paper should talk about the major failure cases and the limitations of their overall method since the reported accuracies even in the simulation are not 100%.**
>
> A: Thank you for the suggestion! We have discussed the failure cases below and updated the manuscript accordingly. We have also included an example in the revised supplementary video.
>
> Failures in simulation: One reason for failures in the simulation results is that we added sensor noises (including camera noises) and perturbations during the training. We found this notably increases the difficulty of optimization, resulting in non-perfect success rate. We have re-trained a policy for the quincuncial environment without these randomizations and the policy does achieve a 100% success rate. We have clarified the training setup for simulation in the revised paper.
>
> Failures in real: The real-world failure cases were mainly caused by the robot stepping too close to the edge of the terrain and slipping into the gap. This is likely due to the discrepancies in the dynamics and vision model. Despite applying domain randomization techniques to mitigate the model discrepancy, we found precisely controlling the landing position of the feet for a moving robot still challenging.
>
>
> [1]. Vision aided dynamic exploration of unstructured terrain with a small-scale quadruped robot. ICRA 2020
>
> [2] DeepGait: Planning and Control of Quadrupedal Gaits using Deep Reinforcement Learning. RA-Letters 2020.

---

### Meta-Review · Area_Chair_LUTg · 2021-08-16

**Recommendation:** Accept (Poster)
**Confidence:** 5

**Metareview:**

The author propose a deep vision based policy to enable a quadrupedal robot to walk on various terrains.  In particular, a RL policy is proposed that takes visual information as input and outputs commands to a low-level MPC controller. Simulation results as well as experimental validation is presented.  The reviewers agree that the paper is well written and presentation is good.  A comparision with other methods would be a good addition to further strengthen the contributions of the paper and establish why the proposed hierarchical policy is better.  Reviewers also point out several related papers that should be part of the literature review and potentially used for comparison.  Please also discuss failure cases.

I am recommending an accept as a poster.

---

> ### Author Response · Authors · 2021-08-30
> **Summary of revisions**
>
> We would like to thank all the reviewers and meta reviewer for the constructive and insightful feedback that help us improve our work further!  ​Based on the reviews, we have made the following revisions to the submission:
>
> - Incorporated missing literatures suggested by the reviewers in the relevant sections.
> - Added failure case discussion and included an example clip of failed trail in the real-world.
> - Added an example where the policy completes the task by adjusting the robot’s heading, demonstrating the turning behavior of our approach.
> - Improved the writing, corrected errors in the text as suggested by the reviewers.
>
> For easy comparison, we have high-lighted the new content in the revised paper in blue color and isolated the new clips in the supplementary material. We have also provided a response for each reviewer to address the individual questions and concerns.

---

### Decision · Program_Chairs · 2021-09-13

**Decision:**

Accept (Poster)

**Comment:**

The author propose a deep vision based policy to enable a quadrupedal robot to walk on various terrains.  In particular, a RL policy is proposed that takes visual information as input and outputs commands to a low-level MPC controller. Simulation results as well as experimental validation is presented.  The reviewers agree that the paper is well written and presentation is good.  A comparision with other methods would be a good addition to further strengthen the contributions of the paper and establish why the proposed hierarchical policy is better.  Reviewers also point out several related papers that should be part of the literature review and potentially used for comparison.  Please also discuss failure cases.

I am recommending an accept as a poster.